# Detection of the Lateral Thermal Spread during Bipolar Vessel Sealing in an Ex Vivo Model—Preliminary Results

**DOI:** 10.3390/diagnostics12051217

**Published:** 2022-05-12

**Authors:** Andreas Kirschbaum, Jan Jonas, Thomas M. Surowiec, Anika Pehl, Nikolas Mirow

**Affiliations:** 1Clinic for Visceral, Thoracic and Vascular Surgery, University Hospital Gießen and Marburg (UKGM), 35043 Marburg, Germany; 2Clinic for Internal Medicine, Diakoniekrankenhaus Freiburg, 79106 Freiburg, Germany; janjonas@web.de; 3Mathematical Optimization, Department of Mathematics and Computer Science, Philipps-University of Marburg, 35037 Marburg, Germany; surowiec@mathematik.uni-marbrug.de; 4Institute of Pathology, University Hospital Gießen and Marburg (UKGM), 35043 Marburg, Germany; anika.pehl@uk-gm.de; 5Department of Cardiac Surgery, University Hospital Gießen and Marburg (UKGM), 35043 Marburg, Germany; nikolas.mirow@t-online.de

**Keywords:** bipolar sealing, lateral thermal spread, thermography, thermal necrosis

## Abstract

Background: As an unwanted side effect, lateral thermal expansion in bipolar tissue sealing may lead to collateral tissue damage. Materials and Methods: Our investigations were carried out on an ex vivo model of porcine carotid arteries. Lateral thermal expansion was measured and a calculated index, based on thermographic recording and histologic examination, was designed to describe the risk of tissue damage. Results: For instrument 1, the mean extent of the critical zone > 50 °C was 2315 ± 509.2 µm above and 1700 ± 331.3 µm below the branches. The width of the necrosis zone was 412.5 ± 79.0 µm above and 426.7 ± 100.7µm below the branches. For instrument 2, the mean extent of the zone > 50 °C was 2032 ± 592.4 µm above and 1182 ± 386.9 µm below the branches. The width of the necrosis zone was 642.6 ± 158.2 µm above and 645.3 ± 111.9 µm below the branches. Our risk index indicated a low risk of damage for instrument 1 and a moderate to high risk for instrument 2. Conclusion: Thermography is a suitable method to estimate lateral heat propagation, and a validated risk index may lead to improved surgical handling.

## 1. Introduction

Bipolar sealing technology has been used successfully in many surgical disciplines for several years [1]. The steadily growing spectrum includes ENT [2,3,4,5,6], visceral surgery, [7,8,9,10,11,12,13,14,15], and gynecology [16,17,18,19,20,21]. Bipolar instruments are used for the preparation, sealing, and severing of vessels, where their use has considerable advantages over conventional techniques. Several studies have shown [5,10,22,23] that intraoperative blood loss is reduced, the overview of the surgical field is improved, and the duration of operations is shortened without increasing postoperative complication rates. In almost all surgical disciplines bipolar sealing technology has therefore largely replaced conventional ligation of blood vessels [22,24,25].

Clinical application is carefully prepared using the appropriate surgical instruments. Many such instruments, for example, Overholt-clamps, are constructed with curved branches and blunt tips in order to avoid accidental tissue damage [16]. After preparation of the vessel to be severed, it is grasped by the instrument branches. In this, care must be taken to grasp the vessel in its entire circumference. The length of the instrument branches therefore often limits the diameter of the vessel to be severed. The instrument branches are closed and the vessel is compressed with a defined and constant pressure. By operating a release switch positioned at the instrument´s handle, the actual sealing process is started. During the sealing process, electrical current passes through the branches, heating the compressed tissue. Tissue resistance between the branches is continuously measured and controlled via a specific algorithm. Energy application is terminated at a predefined level of tissue impedance. At this stage, the vessel is mechanically cut by a blade integrated into the instrument, and the instrument´s branches are opened. The result is carefully inspected by the surgeon, and the surgical process of tissue preparation is continued.

Regarding tissue effects during the sealing process, thermal energy spreads from both sides of the closed instrument branches. This process is known as lateral thermal spread [26]. The level of temperature in the tissue decreases from the instrument branches towards the periphery. Anatomical structures located near the instrument are affected and may become injured to varying extents [27]. Often, the exteriors of the instruments´ branches will have some sort of insulation, so that this effect is mainly observed adjacent to where the branches come together. This lateral heat propagation is difficult to control and presently impossible to avoid. In the literature [28,29,30,31], various organ injuries (recurrent injury in thymectomies, vascular perforations, ureteral injury, and perforations of the intestinal wall) have been described due to lateral thermal expansion. Unfortunately, clinical complications resulting from thermic injury frequently occur after varying time delays. In consequence, knowledge of and due respect for this thermal side effect are of fundamental importance in surgical practice.

An important objective of surgeons and of the manufacturing industry is, therefore, to minimize lateral heat propagation as much as possible [26,32]. Some authors have postulated [29,33] that, at tissue temperatures of 50 °C or more, both reversible and irreversible (necrosis) tissue damage can occur. This view is supported by observations where membrane loosening and tissue edema have been described after laser application [34,35,36]. However, not only the temperature level but also the exposure time and the instrument configuration are significant in this respect [37,38].

In our context, it would be desirable or even required to have a diagnostic tool, which is able to reliably estimate the effect of heat spreading to the surrounding tissue.

The aim of the authors in this project was to examine whether the extent of the potentially critical zone above 50 °C could be determined by a thermographic investigation on an ex vivo vessel model. To address the instrument configuration aspect, the investigations were performed using two different bipolar sealing instruments. In addition, the ratio of reversibly versus irreversibly damaged tissue in the critical zone was analyzed.

This should provide further insight into the risk of damage to surrounding structures during bipolar sealing.

## 2. Materials and Methods

In freshly slaughtered pigs (EU standard 90 kg, male and female), the carotid arteries were taken in their entire length. Both right and left sided vessels were accepted. As the preparations were cadaveric, an animal welfare application was not needed. The vessels were examined on site for possible injuries, and damaged vessels were disposed of immediately. Preparations were packed in moist compresses, cooled, and transported to our laboratory. Transportation time was less than 10 min. Upon arrival, preparations were once more critically inspected. The perivascular connective tissue was removed, so that the vessels became skeletonized. Vessels were repeatedly and carefully checked for injuries, and the diameters of the vessels were determined with the help of a caliper. Only vessels that had diameters between 4 and 5 mm were used for the study. The length of the prepared vessels was approximately 5 cm. Two different commercially available sealing instruments were selected, and the preparations were randomly assigned to experimental groups thus formed.

In group 1, the sealing instrument was marSeal^®^ 5 plus Maryland (KLS Martin, Gebrüder Martin & CoKG, Tuttlingen, Germany). The branches of this instrument have a slightly concave shape with a sealing length of 18 mm. A special coating on the outside is intended to reduce lateral heat propagation. In group 2, BiCision^®^ 5mm (Erbe, Tübingen, Germany) was employed. The BiCision^®^ instrument is different in design and displays a shell shape in cross section. This allows the thermofusion zone to be expanded by drawing in more vessel length. The branches are not specifically insulated.

It is to be emphasized that both industrial devices are certified for clinical use and are regarded as successful instruments.

In total, 15 vascular preparations were included in each group. The carotids were positioned vertically onto our measuring device and fixed with crocodile clips. The respective sealing instrument was attached to a tripod, and the vessel was grasped with the instrument branches. For all measurements, a thermal camera (Optris PI 640, Berlin, Germany, optical resolution 640 × 480) was aligned in a static fixed position for temperature recording at a distance from the object of 20 cm (see Figure 1A). The branches of the sealing instrument were closed, and the sealing process was initiated. Recording of the thermal propagation was started and was continued until the respective sealing process was completed (see Figure 1B). The instrument branches were opened, the sealed vessel segment was spread out onto a silicone block and placed in a jar containing a solution of 10% formaldehyde.

The recorded temperatures were entered into a specifically designed MATLAB (The MathWorks Inc., Natick, MA, USA) program and printed out as a graph. This allowed for exact identification of the zone above 50 °C (see Figure 2).

Histological processing was performed by experienced staff at the Institute of Pathology, University of Marburg. Preparations were cut perpendicular to the sealing plane and stained with hematoxylin–eosin according to the standard scheme (see Figure 3). Histological sections were digitized, and the extent of the respective tissue necrosis was marked by an experienced pathologist. From a reference point marking the edge of the branches, 15 measurements of the extent of necrosis were made. Means and standard deviations were determined separately for tissue areas above and below the instrument branches. By referring to the thermographic records, we were able to precisely quantify the temperature at the margins of the necrosis zone. The distribution of measured values was not normal; so, a nonparametric test was employed for statistical evaluation. The individual groups were tested for significance by a nonparametric Mann-Whitney U test (level of significance *p* < 0.05).

To assess the risk of damage during the sealing process due to lateral thermal expansion, we defined a Risk Index of Lateral Thermal Expansion (RILATE):$$\begin{array}{c} Risk\;index\;of\;the\;lateral\;thermal\;expansion\;\left(RILATE\right)\\ \;\;\;\;\;\;\;\;\;\;\;\;\;\;\;\;\;\;\;\;\;\;\;\;\;\;\;\;=\left(Expansion\;of\;the\;necrotic\;zone:Total\;expansion\;of\;the\;\mathrm{critical}\;\mathrm{zone}\left(>50{}^{\circ}C\right)\right)\times 100 \end{array}$$

This index refers to the extent of the necrosis zone in relation to the total extent of the potential critical zone >50 °C. It was classified as low when ≤30%, moderate between 31 and 60%, and high ≥61%. According to our estimate, the risk of injury to the tissue observed was increased if an elevated RILATE-Index was calculated.

## 3. Results

In sealing with the marSeal^®^ instrument, the expansion of the critical zone above the instrument branches at above 50 °C was calculated as a mean of 2315 ± 509.2 µm, and the mean necrotic expansion was 412.5 ± 79.0 µm. The temperature at the margin of the necrosis zone was 64.93 ± 4.1 °C. Thus, an index of 17.8% was calculated. Looking at the situation below the instrument branches, the critical zone extended to 1700 ± 331.3 µm, and the necrosis zone was 426.7 ± 100.7 µm. The temperature at the margin of the necrosis zone was 63.42 ± 3.38 °C. The corresponding index amounted to 25.1%. The extents of the zones upwards and downwards of the instrument branches differed significantly (*p* = 0.0006), but there was no significant difference between the extents of the necrosis zones (*p* = 0.98). Temperatures at the margins of each necrosis zone were not significantly different in comparison (*p* = 0.28). Due to the significantly larger extent of the critical zone, there was also a significant difference in the indices (*p* = 0.05).

For the Bicision^®^ instrument, the extent of the critical zone above the instrument branches was determined to be 2032 ± 592.4 µm and the necrosis zone was 642.6 ± 158.2 µm. The temperature at the margin of the necrosis zone towards the periphery was measured to be 60.42 ± 4.2 °C. An index of 31.62% was calculated. Below the instrument branches, an extension of the critical zone of 1182 ± 386.9 µm and a necrosis zone of 645.3 ± 111.9 µm were measured. The associated temperature at the margin of the necrosis zone was 55.28 ± 4.1 °C. An index of 54.59% was calculated. The extents of the critical zone above and below the instrument branches differed significantly (*p* < 0.0001). In contrast, the actual necrosis zones above and below did not differ significantly (*p* = 0.9). The temperature at the end of necrosis above the instrument branch was significantly higher than below (*p* = 0.002). There was a significant difference in the calculated indices above versus below the instrument branch (*p* < 0.0001). Table 1 provides an overview. Histological examination of all sections showed a sharp transition between the necrosis zones and the areas of only little tissue alteration. This area mostly showed only discrete loosening of the tissue as a result of thermal exposure.

## 4. Discussion

When using a bipolar sealing instrument, the lateral heat propagation should be as low as possible in order not to damage the surrounding tissue. In particular, irreversible unwanted tissue damage must certainly be avoided. In a clinical setting, collateral damage during the procedure to tissue may go unnoticed by the surgeon. Clinical complications may only come with a delay of several days. This aspect plays a major role in any new instrument development. In this context, lateral heat propagation depends on the level of the temperature, duration of tissue exposure, and on the technical configuration of the instrument branches. The focus of our study was to determine whether thermography was able to identify and determine lateral heat propagation during the use of sealing instruments. An important aspect was that this should be possible using instruments of different technical construction. Our thermographic measurements confirmed the conjectures of [33], that tissue damage during bipolar carotid seals occurs at tissue temperatures >50 °C. We consider this region as the potential critical zone. By means of a thermographic recording, heat distribution in bipolar sealing is measured in real time, accurately and reliably. The use of thermography does not require great effort. The technology can be used in many different settings and it allows for long-term measurements. In our opinion, it is clearly superior to methods reported in the literature [29,39] such as a measurement via a probe or histological examination alone [29].

Thermography can be used to determine the temperature at variable distances from the instrument branches. With commercially available temperature probes that are inserted into the tissue, temperatures can only be measured at specific points. In comparison to thermography, this is a major disadvantage.

However, as the camera must be securely positioned, cannot be moved during the measurements, and has to be precisely aligned with the target object, its use in a clinical setting may be a difficult task. Other means are needed for risk reduction.

Identification of critical zones seems crucial, and for their further assessment we performed histological examinations. All sections displayed a defined transition between the necrosis zone and little-altered tissue. Differing from some published statements [33,37], we found that temperatures much higher than 50 °C were sometimes tolerated before tissue destruction was observed. Other authors reported similar conclusions [29,35].

There were limitations to our model on various grounds. Histological examination, which is considered the most reliable and best-established method to assess thermal necrosis, is nevertheless of limited value in this study. It documents the biological state of the tissue examined at a specific defined moment in time. It cannot always predict further development, i.e., tissue from the “reversibly damaged” zone may either recover to a viable state or become fully necrotic.

A further obstacle to evaluation of tissue damage is the fact that exact assessment of histological sections is a challenge even for experienced pathologists. The tissue blocks must be cut with great precision. In addition to section artifacts, accurate quantitative measurement of the necrotic zones and especially of the transition zones requires advanced skills.

Furthermore, in our experiments neither in the sealing zone nor in the adjacent parts were the vessels to be sealed perfused with blood. We cannot, therefore, provide information on the effect of heat energy on filled vessels. Nor can we provide information on its effect on surrounding tissue other than blood vessels, as in our model prior to the sealing process all perivascular connective tissue was removed. In a clinical setting, this may not always be possible, and tissue reactions to heat propagation may be different. We are convinced that our model can contribute to a better understanding of blood vessel sealing and corresponding lateral heat propagation though.

In order to gain practical information regarding the sealing procedure, we formed a new index that estimates the risk of lateral tissue damage. Risk levels were defined, and we consider them to reflect the experimental results [40]. The higher the index value calculated, the higher the expected risk of irreversible damage to surrounding tissue caused by the sealing process. This provides instrument developers and users with an important criterion for assessing the significance of lateral heat propagation in a sealing instrument.

The index may therefore contribute to more careful surgical use of a defined instrument but may also help to improve future instrument design.

To further substantiate our results and to gain further insight into the matter in vivo animal experiments may be a next step. It is most important to determine how blood filled vessels will influence the extent of lateral heat propagation of tissue adjacent to the sealing area. A thermal camera could be installed and used in conditions closer to clinical reality. The focus could be on improved control of surgical handling and—ideally—on advancing the technical design of sealing instruments.

## 5. Conclusions

Thermography is well suited to record lateral heat propagation during bipolar sealing. The essential information provided was determination of the extent of the temperature zone above 50 °C.

Surgeons and industry should assess the exact risk of potential damage. Surgical handling should be further adapted to anatomical details and to specific technical properties of instruments employed. Thermic isolation of the instrument branches may help in reducing lateral heat propagation. The risk index presented in this study may assist in quantifying and classifying the relevance of lateral thermal expansion in bipolar blood vessel sealing.

## Figures and Tables

**Figure 1 diagnostics-12-01217-f001:**
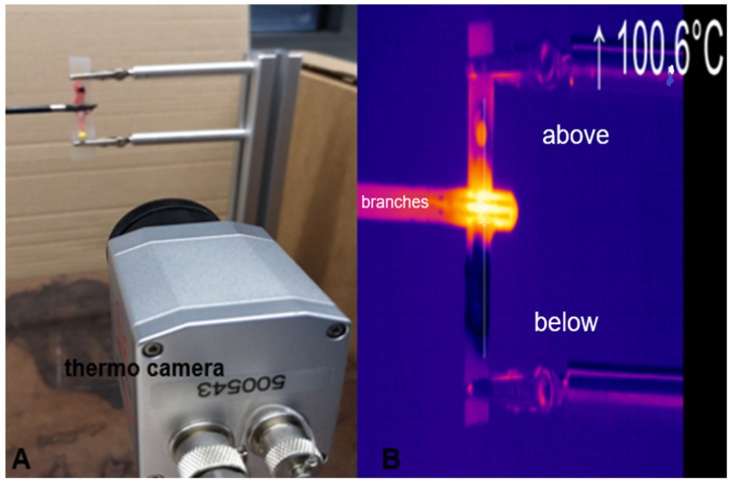
(**A**): Experimental setup, with the position of the thermal camera towards the sealed vessel (**B**): Example of a measurement by thermal camera (own pictures).

**Figure 2 diagnostics-12-01217-f002:**
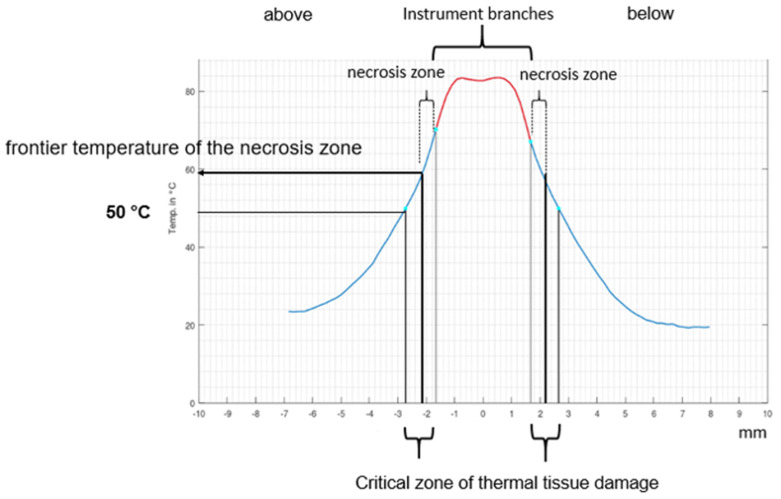
Example of a temperature curve, demonstrating the different zones for necrosis and potential thermal damage (print from the MATLAB program).

**Figure 3 diagnostics-12-01217-f003:**
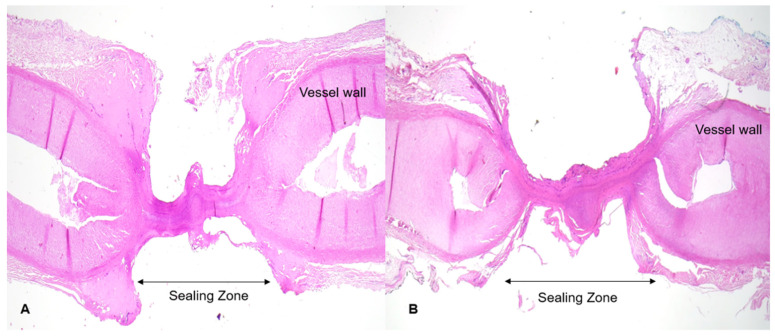
(**A**) Vessel just sealed by marSeal 5 plus (**B**) Vessel sealed by BiCision (HE staining) (magnification 4.5×).

**Table 1 diagnostics-12-01217-t001:** Overview of the expansion of the critical zone, the necrosis zone, the height of the temperature at the end of the necrosis zone, and the indices (*n* = 15 per group) for the instruments marSeal^®^ 5 plus and BiCision^®^.

marSeal^®^	Critical Zone above (µm)	Necrosis Zone (µm)	Frontier Temperature (°C)	Rilate
1	2070	421.9	57.5	20.3
2	2830	442.6	66.5	15.6
3	2630	439.7	66.0	16.7
4	2570	535.3	68.7	20.8
5	2660	461.5	64.7	17.3
6	1900	285.7	63.5	15.0
7	1360	327.8	59.4	24.1
8	1620	385.1	63.0	23.7
9	2500	316.2	71.3	12.6
10	2630	373.5	67.5	14.2
11	1830	486.8	68.3	26.6
12	2050	521.7	58.2	25.4
13	3310	492.8	64.7	15.7
14	2290	374.6	70.1	16.3
15	2470	322.8	64.6	13.1
mean	2315	412.5	64.93	17.8
SD	509.2	79.0	4.1	4.6
	**Critical Zone below (µm)**	**Necrosis Zone (µm)**	**Frontier Temperature (°C)**	**Rilate**
1	1990	416.2	59.6	20.9
2	1580	497.6	65.1	31.5
3	1540	385.9	63.1	25.0
4	1590	562.2	60.7	35.4
5	1220	606.8	57.9	49.7
6	1590	384.4	61.3	24.1
7	2030	522.7	66.8	25.7
8	1430	494.5	62.1	34.5
9	1980	332.6	67.5	16.8
10	1660	331.7	62.6	19.9
11	2480	407.1	70.5	16.4
12	1330	514.4	59.9	38.6
13	1520	373.2	64.6	24.5
14	2010	295.7	65.8	14.7
15	1550	275.6	63.8	17.8
mean	**1700**	**426.7**	**63.42**	**25.1**
SD	**331.3**	**100.7**	**3.38**	**9.8**
**BiCision^®^**	**Critical Zone above (µm)**	**Necrosis Zone (µm)**	**Frontier Temperature (°C)**	**Rilate**
1	1600	886.6	56.5	55.4
2	1480	733.8	56.0	49.6
3	2750	956.8	65.2	34.8
4	1950	654.8	58.7	33.6
5	1760	479.1	57.9	27.2
6	1650	518.4	57.3	31.4
7	2600	497.9	64.0	19.2
8	1560	654.8	60.6	42.0
9	2270	768.1	60.9	33.8
10	2340	597.0	61.7	25.5
11	1730	499.5	59.3	28.9
12	2970	567.0	64.6	19.0
13	1400	582.9	55.4	41.6
14	3090	811.1	70.6	26.2
15	1330	431.5	57.6	33.4
mean	2032	642.6	60.42	31.62
SD	592.4	158.2	4.2	10.3
	**Critical Zone below (µm)**	**Necrosis Zone (µm)**	**Frontier Temperature (°C)**	**Rilate**
1	1270	694.7	52	54.7
2	1440	688.3	56.6	47.8
3	2010	973.6	51.2	48.4
4	730	618.2	52.1	84.7
5	720	535.4	52	74.4
6	1530	542.8	59.8	34.5
7	1490	596.4	64.7	40.0
8	820	603.1	54.2	73.5
9	1280	696.9	54.7	54.4
10	920	556.8	53.8	60.5
11	820	592.8	53.3	72.3
12	890	588.5	53	66.1
13	870	748.8	51.3	86.1
14	1550	682.3	62.4	44.0
15	1390	561.3	58.1	40.4
mean	1182	645.3	55.28	54.59
SD	386.9	111.9	4.1	16.8

SD = Standard Deviation.

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
