# Peer review of "Detection of the Lateral Thermal Spread during Bipolar Vessel Sealing in an Ex Vivo Model—Preliminary Results"

_diagnostics, 2022, doi:10.3390/diagnostics12051217_

Round 1
Reviewer 1 Report
I think your paper is well written and interesting data are provided.
Some important limits have been shown and correctly acknowledged (eg: no information about the efficacy of sealing).
I think your model may only partially reproduce the clinical practice; as a further possibile development in new studies, I would rather suggest not to test the instruments only on arteries (consider other tissues like fat, muscles, connettive that are much more frequently dissected and sealed during surgery; consider venous segments; consider arteries without dissecting perivascular tissues; consider different experiments with very different durations of application of the energy).
I would therefore recommend your paper for publication by adding in the title a clear reference to preliminary results.
Author Response
I think your paper is well written and interesting data are provided.
Some important limits have been shown and correctly acknowledged (eg: no information about the efficacy of sealing).
I think your model may only partially reproduce the clinical practice; as a further possibile development in new studies, I would rather suggest not to test the instruments only on arteries (consider other tissues like fat, muscles, connettive that are much more frequently dissected and sealed during surgery; consider venous segments; consider arteries without dissecting perivascular tissues; consider different experiments with very different durations of application of the energy).
I would therefore recommend your paper for publication by adding in the title a clear reference to preliminary results.
Answer to the Reviewer: Thank you very much for the review of our manuscript. The aim of this work was to develop an appropriate model in order to describe lateral heat propagation initially staring with carotid arteries and including a thermal camera. We agree with you that in surgical practice other tissues are subject to bipolar sealing. In further studies planned, lateral heat propagation in different tissues will also be investigated by changing the energy output.
The title was altered according to your recommendation, “preliminary results” were included.
Reviewer 2 Report
The paper is very good and should be published, but there are some points that can be discussed and we list them:
1.The first is that cadaveric bloodless vessels were used which is something the authors themselves point out in the discussion and it is expected that they will not behave similarly to blood filled vessels. Although it seems reasonable to use them, however, no one can confidently claim that the behavior of blood filled vessels will be the same as that of cadaveric vessels.
2.We would also like to see more histological images of at least two of both types of sealing instruments (marSeal® and BiCision®),although we do not consider it necessary for the publication to make the correction.
3.We would still prefer to see the whole data table as it could (due to the fact that there were only 30 vessels 15 + 15) be included in the paper for completeness and not just see the statistical analyses with significances.
Where we think there should be a change, however, is in the abstract where thermography is referred to as an instrument while it is a method. ( thermal camera is the instrument).
Author Response
The paper is very good and should be published, but there are some points that can be discussed and we list them:
1.The first is that cadaveric bloodless vessels were used which is something the authors themselves point out in the discussion and it is expected that they will not behave similarly to blood filled vessels. Although it seems reasonable to use them, however, no one can confidently claim that the behavior of blood filled vessels will be the same as that of cadaveric vessels.
Answer to the Reviewer: We consider the “bloodless vessels” model suitable to initially investigate essential effects of bipolar sealing. However, it must be assumed that these effects are partially different in blood perfused vessels. In the specific setting the authors know of no reliable data for comparison in this regard. However, it must be considered that after closing the instrument branches, the vessel is compressed and thus the blood flow is completely interrupted in the sealing area. The actual sealing process then takes place in a vessel area that is not perfused. Of course this will not apply to the proximal and distal parts of the vessel next to the sealing area.
2.We would also like to see more histological images of at least two of both types of sealing instruments (marSeal® and BiCision®),although we do not consider it necessary for the publication to make the correction.
Answer to the Reviewer: Figure 3 shows an exemplary histological image of a sealing zone using the marSeal instrument to demonstrate the effect of vessel sealing. In addition, we have added a histological image after sealing with the BiCision instrument.
3.We would still prefer to see the whole data table as it could (due to the fact that there were only 30 vessels 15 + 15) be included in the paper for completeness and not just see the statistical analyses with significances.
Answer to the Reviewer: We added the specific data requested to the table for completeness.
Where we think there should be a change, however, is in the abstract where thermography is referred to as an instrument while it is a method. (thermal camera is the instrument).
Answer to the Reviewer: We changed the abstract accordingly.